# Immune Resistance and EGFR Antagonists in Colorectal Cancer

**DOI:** 10.3390/cancers11081089

**Published:** 2019-07-31

**Authors:** Guido Giordano, Andrea Remo, Almudena Porras, Massimo Pancione

**Affiliations:** 1U.O.C. Medical Oncology, Ospedali Riuniti, Azienda Ospedaliero Universitaria, 251 Foggia, Italy; 2Pathology Unit, Mater Salutis Hospital AULSS9, “Scaligera”, 37122 Verona, Italy; 3Department of Biochemistry and Molecular Biology, Faculty of Pharmacy, Complutense University Madrid, 28040 Madrid, Spain; 4Health Research Institute of the Hospital Clínico San Carlos (IdISSC), 28040 Madrid, Spain; 5Department of Sciences and Technologies, University of Sannio, 82100 Benevento, Italy

**Keywords:** colorectal cancer, EGFR, resistance, targeted therapies, immune microenvironment

## Abstract

Targeting the epidermal growth factor receptor (EGFR) either alone or in combination with chemotherapy in patients with RAS wild type metastatic colorectal cancer (mCRC) has revolutionized the treatment of CRC, but with less results than initially envisaged. In recent years, the discovery of multiple pathways leading to the escape from anti-EGFR therapy has revealed an enormous complexity and heterogeneity of human CRC due to the intrinsic genomic instability and immune/cancer cell interaction. Therefore, understanding the mechanistic basis of acquired resistance to targeted therapies represents a major challenge to improve the clinical outcomes of patients with CRC. The latest findings strongly suggest that complex molecular alterations coupled with changes of the immune tumor microenvironment may substantially contribute to the clinical efficacy of EGFR antagonist. In this review, we discuss the most recent findings that contribute to both primary and acquired anti-EGFR therapy resistance. In addition, we analyze how strategies aiming to enhance the favorable effects in the tumor microenvironment may contribute to overcome resistance to EGFR therapies.

## 1. Introduction

Colorectal cancer (CRC) is one of the most common types of cancers in humans and is closely linked to the global cancer-related mortalities worldwide [1]. The global burden of colorectal cancer is expected to increase by 60%, to more than 2.2 million new cases and 1.1 million deaths by 2030 [1,2]. Around three-quarters of the CRC patients present metastases at the time of diagnosis and need systemic therapies [3]. Depending on the tumor characteristics, extend of disease and genetic alterations and other factors, the treatment of patients with metastatic CRC varies. Since 1995, treatment regimens have included capecitabine, irinotecan, oxaliplatin, bevacizumab, cetuximab, panitumumab, aflibercept, and reforafenib [3,4]. These treatments have doubled the median survival of patients and improved the 5-year survival from less than 1% to 20%. Despite advances in precision oncology, little progress has been made in metastatic colorectal cancer (mCRC) in recent years. In late stage CRC, the most commonly used targeted therapies are monoclonal antibodies (mAbs) cetuximab and panitumumab, which inactivate EGFR1 signaling. Unfortunately, they are only effective in a small percentage of patients [5,6,7] due to primary or secondary/acquired resistance to this kind of therapy. Notably, even the patients that initially respond to the EGFR blockade therapy usually develop secondary resistance over time by a number of causes [8,9,10,11,12,13,14]. Alterations in RAS, among other genes, represents the main mechanisms of this resistance [11,12]. Despite these efforts, additional mechanisms of resistance to EGFR blockade are thought to be present in CRC and little is known about the determinants of sensitivity to this therapy [13,14,15,16]. Among them, it is important to mention *Raf*, *PI3K*, and *MAP2K1* mutations, *ERBB2* and *MET* amplification, low EGFR gene copy number, IGF1 overexpression, *PTEN* loss, over-activation of STAT3 by JAK, or mediated by nuclear PKM2 [5,6,9,10]. The latest evidences strongly suggest that complex molecular alterations coupled with changes in the tumor microenvironment may substantially contribute to the clinical efficacy of EGFR antagonists, even in CRC that are both *KRAS* and *NRAS* wild type [5,6,7,8,9,10,14]. In this review, we first discuss the mechanisms by which EGFR secondary alterations induce compensatory functions against the inhibition of EGFR. Second, we analyze how systemic treatment of CRC might result in altered tumor microenvironment and acquired resistance to anti-EGFR therapy. Lastly, we discuss the potential strategies aiming to enhance the favorable effects in the tumor microenvironment to overcome resistance to EGFR-based therapies.

## 2. The Roles of EGFR/ERBB Therapy in Colorectal Cancer

The epidermal growth factor receptor (EGFR) family of proteins is formed by four receptor tyrosine kinases, EGFR/ERBB1, ERBB2/HER2, ERBB3/HER3, and ERBB4/HER4. They are generally activated by the binding of their ligands (EGF, transforming growth factor-α (TGF-α) and others) to the extracellular domain of EGFR, HER3, and/or HER4 leading to their homo- and/or heterodimerization with other EGFR family members [17,18]. In cancer, these ligands come from either tumor or stromal cells, acting in an autocrine or paracrine way, respectively. Once the ligand binds to its receptor, a conformational change is produced and its tyrosine kinase is activated leading to the stimulation of several downstream signaling cascades. Among the different pathways activated, it is important to highlight the RAS/RAF/ERKs, PI3K/AKT/mTORC1, SRC, PLCγ/PKC, p38 MAPK, JNKs, and JAK/STAT pathways, which contribute to regulate cell proliferation, survival, apoptosis, and other cellular functions [14,17,18].

Not all the EGFR family members present the same properties. For example, HER2 is considered an orphan receptor, with no known ligands for it and HER3 is characterized by its low intrinsic tyrosine kinase activity [19]. However, when HER3 forms hetero-dimers with HER2, several downstream signaling pathways are activated in different types of cancer cells. On the other hand, when HER2 is overexpressed in tumor cells, it is able to form homodimers [19]. The genes from EGFR family are relevant in cancer. Hence, several studies developed during the last decades has revealed the presence of different genetic alterations (e.g., mutations, amplification, etc.) in members of this family of receptors in a great variety of tumors, including colorectal cancer. This has been associated with tumor initiation and progression as well as with a poor prognosis [19,20]. Therefore, EGFR-targeted inhibitors, including tyrosine kinase inhibitors (TKIs) and monoclonal antibodies (mAbs) have been developed for its use in the treatment of several cancers. The combination of the classic chemotherapy with antibodies against EGFR or VEGFR (to inhibit angiogenesis) has contributed to increase the overall survival in metastatic colorectal cancer (mCRC). However, the benefits of using the TKIs are unclear, which represents another potential relevant targeted therapy [4].

Recently, the molecular profile has allowed CRC stratification into different consensus molecular subtypes (CMS1-4), which show great potential as predictive biomarkers for the efficacy of conventional and targeted treatments [21,22]. Note that, given the broad scope of the material covered and the space constraints, readers are invited to consult dedicated reviews on these topics [21,22,23]. Preclinical and clinical studies have revealed a highly complex relationship between tumor biology and the efficacy of the anti-EGFR therapy. Primary resistance, clinical relapse, and metastasis in response to anti-EGFR therapy is common in CRC. Besides *KRAS* mutation, the latest findings suggest that a growing number of oncogenic alterations or specific polymorphic variants in EGFR or genes that regulate its turnover or immune responses to anti-EGFR mAbs might be associated with the clinical response to EGFR based therapies (Table 1 and Table 2) [24,25,26,27,28,29,30]. These studies raise the important question of how EGFR-targeted may be exploited to evoke more effective treatment outcomes. Although anti-EGFR mAbs have shown a discrete success against metastatic CRC bearing wild-type KRAS, it remains difficult to cure patients, even when anti-EGFR are combined with chemotherapy, because tumors inevitably develop an acquired resistance.

## 3. Acquired Resistance to Anti-EGFR Treatment in CRC Patients

Most CRC patients bearing wild-type (wt) *KRAS* initially respond to the treatment with anti-EGFR monoclonal antibodies, cetuximab and panitumumab [3]. However, after 3–12 months they become resistant to the treatment (Table 3) and new genetic alterations arise, 50% of them being *KRAS* mutations [31]. This has been found not only in preclinical models, but also in biopsies (from primary and metastatic tumors) and circulation tumor DNA (ctDNA) derived from patients [5,31,32]. These mutated *KRAS* clones that emerge in blood during anti-EGFR treatment, decline upon withdrawal of the specific antibodies, which leads to the recovery of drug sensitivity [31,32].

It is important to mention that among the different *KRAS* mutations, KRASQ61 is the one that is often associated with the acquired resistance to EGFR blockade, most likely because it functions even when protein expression is low [31]. In contrast, KRASG12 mutation appears to require an upregulation of translation to promote resistance.

It is not totally clear if these genetic alterations induced by a prolonged EGFR blockade are new mutations or they are already present in some intra-tumor subclones. According to the heterogeneity of tumors and based on in vitro and in vivo studies, it is more likely that a clonal selection of pre-existing resistant cells may occur along the treatment [12,14]. Therefore, the resistant cells will be selected during treatment.

The emergence of secondary mutations in *NRAS* and *BRAF* or EGFR is also associated with acquired resistance to anti-EGFR treatment, although these mutations are rare as compared to those in *KRAS* [10,34]. In some cases, concomitant mutations in *KRAS* and *NRAS* were also detected in the plasma of these CRC patients. Recent data indicate that a patient bearing wild-type KRAS and NRAS initially responded to the combination of chemotherapy and cetuximab, but within the time a K57T MEK1 new mutation, not present before the treatment with cetuximab, was detected in the primary tumor and in liver metastases [34,35]. This suggests that alterations in *MEK1* gene can represent a novel mechanism of cetuximab-induced resistance. As a consequence, the use of the MEK inhibitor, trametinib, in combination with the previous therapy was able to re-sensitize cells to anti-EGFR antibodies, at least, in vitro. However, although the treatment with trametinib and panitumumab reduced liver metastases, the growth of other metastases was not prevented [34,35].

A number of studies revealed the presence of HER2 amplification and/or activation by overexpression of the HER3/4 ligand, heregulin, in CRC patient samples or patient-derived xenograft models with acquired resistance to anti-EGFR treatment [36,37,38,48]. Among the EGFR family, HER2 is the one with the strongest catalytic activity and it usually dimerizes with EGFR (HER1) and HER3 [17]. Therefore, a high expression of heregulin leads to HER2 activation as well as that of downstream signaling pathways such as ERKs. In addition, when overexpressed, HER2 leads to a prolonged ERKs activation and an increase in HER3 and PI3K/Akt activation [17,36]. Therefore, the combination of HER2 and EGFR inhibitors is able to decrease the in vitro growth of cetuximab-resistant CRC cells and to induce in vivo regression of tumors generated in animal models [17,19]. This indicates that the combination of an anti-HER2 antibody, pertuzumab (a humanized IgG1 monoclonal antibody that interferes with HER2 dimerization) with inhibitors of EGFR and HER2 tyrosine kinases would improve the treatment of patients showing cetuximab resistance. The HERACLES trial has confirmed the effectiveness of dual blockage of HER2 with trastuzumab plus lapatinib in patients with heavily pretreated HER2-positive mCRC, refractory to the anti-EGFR antibodies cetuximab or panitumumab. Furthermore, combination treatment with pertuzumab and lapatinib (a small molecule, inhibitor of EGFR and HER2 tyrosine kinase) achieved a better tumor control than either agent alone [36,37,38,48]. However, it is highly debated whether re-activation of these pathways are critical for colorectal tumor survival in a clinical setting [22].

Although with a low frequency, EGFR can be mutated upon prolonged treatment with anti-EGFR therapy. In particular, a mutation in the extracellular domain of EGFR (S492R) was found in some CRC patients with acquired resistance to cetuximab [19,21,33]. Because of the substitution of serine by arginine at position 492, the affinity of EGFR for its ligand decreases, which also interferes with cetuximab binding. Other mutations in EGFR has been also detected in ctDNA from CRC patients with acquired resistance to anti-EGFR therapy, such as EGFR (S464L) or EGFR (G465R) [19,21]. Recently, the genomic profiling of colorectal cancer in ctDNA has revealed an exceptional number of genetic variants in the blood samples from advanced stage CRC patients [46]. Notably, individual patients exhibited multiple genes of resistance to anti-EGFR antibody [19,21,46]. With the aim of getting a better understanding of the mechanisms of resistance to HER2-directed and EGFR-directed therapies, it has been recently reported that increased expression of *MUC1* and *MET***,** decreased expression of *PTEN*, and an activating mutation in *PIK3CA* is accompanied with *HER2*-amplified in mCRC. These findings strongly highlight the molecular heterogeneity of the resistance to anti-EGFR therapy [39,46,49,50,51,52].

The analysis of tumor tissues from CRC patients who developed resistance to the treatment with anti-EGFR antibodies evidenced the presence of *MET* amplication in around 40% of them [53]. *MET* amplification was also detected in ctDNA from CRC patients with acquired resistance to anti-EGFR therapy [52,53]. In addition, overexpression of TGF-α can lead to MET activation acting through EGFR-MET heterodimer, which contributes to the cetuximab resistance in colorectal cancer cells [47,53]. All these data support a role for MET in the generation of an acquired resistance to cetuximab in CRC patients. According to this, in vitro studies demonstrated that HGF-mediated Met activation prevented cetuximab-induced apoptosis or cell cycle arrest in CRC cells [39,51,52]. Preclinical and clinical data suggest that patients with amplified *MET* in tumors benefit from MET-targeted therapy. In addition, c-MET overexpression, irrespective of primary sites or molecular markers, predicts a shorter PFS during bevacizumab treatment in patients with CRC [53]. A clinical trial targeting amplified *MET* in metastatic CRC is currently underway [53].

The increase in the levels and/or activation of other tyrosine kinase receptors such as IGF1R or VEGFR, as well as the activation of EGFR downstream pathways can be also involved in the generation of secondary resistance to EGFR blockade [47]. ERKs and PI3K pathways are relevant EGFR downstream signaling cascades, shared by other tyrosine kinase receptors. Its activation in response to anti-EGFR antibodies can contribute to generate resistance [47]. Therefore, a strategy to improve CRC patients’ treatment would be based on the combination of an anti-EGFR therapy and MEK or PI3K/Akt inhibitors. Recently, the presence of high levels of EphA2 tyrosine kinase receptor has also been associated with CRC progression and resistance to the treatment with FOLFIRI and cetuximab, as well as with a poor prognosis. As a consequence, the treatment with a specific EphA2 inhibitor is able to overcome primary and acquired resistance to anti-EGFR therapy both in vitro and in vivo [54]. The expression and activation of IGF1R can also be increased in anti-EGFR resistant CRC cells [47,54]. In fact, in vitro experiments using different CRC cell lines resistant to a combination of cetuximab and refametinib (a selective MEK-inhibitor) revealed an upregulation of IGF1R mRNA levels and increased phosphorylation of IGF1R, in addition to an increase in different members of EGFR family [47]. PI3K/Akt pathway was also activated and its inhibition reverses the resistance. On the other hand, IGF1R knock-down blocked Akt activation. Therefore, IGF1R will also contribute to the generation of secondary resistance to anti-EGFR therapy in CRC [47,54]. VEGF/VEGFR signaling can also be involved in acquired resistance to cetuximab or gefinitib treatment. In fact, the inhibition of VEGFR2 in xenograft assays resistant to EGFR blockade was able to reduce tumor growth [4,14,23]. Forkhead box class O 3a (FOXO3a) protein was found to be upregulated in CRC tissues from patients with an acquired resistance to cetuximab [43,45]. Moreover, high FOXO3a and p38 MAPK expression can predict the response to cetuximab in patients with CRC harboring wt KRAS (Figure 1) [43,45]. Similarly, CRC cell lines chronically treated with this anti-EGFR therapy also showed an increased expression of FOXO3a. In addition, FOXO3a induced c-Myc transcription led to the expression of genes involved in metabolism such as Pyruvate kinase muscle isozyme M2 (PKM2). The down-regulation of both, FOXO3a and c-Myc, highly reduced the survival and tumor growth. Therefore, FOXO3a will be a key regulator of cetuximab-induced CRC resistance acting through c-Myc [32,43]. At the molecular level, cetuximab activates the transcription factor FOXO3a and promotes its nuclear translocation via p38-mediated phosphorylation, leading to the upregulation of its target genes p27 and Bcl-2-like protein 11 (Bim) and the subsequent induction of apoptosis [32,48]. Accordingly, basal expression of Bim protein is lower in double-mutant KRAS and PIK3CA CRCs. Notably, these double-mutant cancer subtypes undergo apoptosis and tumor growth inhibition following treatment with inhibitors of SRC and the MEK/ERK pathway [32]. These data support the hypothesis that a potentially promising alternative to overcome resistance mechanisms will be to apply a therapy in the upfront setting in order to suppress and ideally eradicate pre-existing resistant clones while they still are present in a low frequency subpopulation.

## 4. Contribution of Tumor Microenvironment to Acquired Resistance to EGFR Blockade

Intratumoral heterogeneity is recognized as a major mechanism underlying treatment failure for molecule-targeted agents (Figure 1) [4,14,44]. CRCs require interaction with many different host immune cell populations for their growth and survival, including regulatory T cells (Treg) and myeloid-derived suppressor cells (MDSC), mast cells, and tumor associated macrophages (TAM) [55]. Recent studies suggest that a substantial portion of the effects attributed to EGFR antagonist treatment may be based on indirect effects beyond cancer cells [40,41,42,55,56,57,58,59,60,61,62].

It is well known that EGFR is expressed in different hematopoietic cell types, including macrophages, monocytes, certain T cell subsets, such as effector CD4 T cells and FoxP3-expressing CD4 T Tregs, being relevant for their function. Therefore, it is likely that EGFR antagonist can interfere in the function of these leukocytes, contributing to the clinical efficacy of anti-tumor treatments (Figure 2) [13,14].

For example, the combination of NK cells and cetuximab has been proposed as a rationale to strengthen NK cell immunotherapy in mCRC patients [59]. Furthermore, increased expression of EGFR can be observed in myeloid cells from the tumor stroma and associates with tumor progression and reduced survival time of patients with gastrointestinal cancers, including colorectal [40,57]. EGFR expression in myeloid cells increases activation of STAT3 and expression of survivin in intestinal epithelial cells and expression of IL6 in colon tissues. Notably, deletion of EGFR from myeloid cells, but not from intestinal epithelial cells, protects mice from colitis-induced intestinal cancer and ApcMin-dependent intestinal tumorigenesis [40]. Therefore, the expression of EGFR by myeloid cells of the colorectal tumor stroma, rather than the cancer cells themselves, also contributes to tumor development [40]. Furthermore, a significant correlation between high EGFR activity in tumor cells and macrophage-tumor cell proximity was found to partially account for the intratumoral heterogeneity in EGFR activity observed in CRC [32]. Anti-EGFR and oxaliplatin based chemotherapy powerfully induce CD8+ cells mobilization within the metastatic site in wt RAS CRC patients, supporting a role for the immune response in mCRC under RAS status dependence [58]. This also agrees with the fact that tumors responsive to EGFR antagonist treatment in vivo are often not sensitive to monoclonal antibody treatment in cell culture when explanted, where the immune cells are not present. In fact, inhibition of EGFR signaling in colon cancer cells modulates cytokines and growth factors secretion (e.g., IGF-1) and prevents M1-to-M2 macrophage polarization within the tumor, thereby inhibiting cancer cell growth [40,58]. In addition, there are also evidences that adjuvant chemotherapeutic agents employed for CRC treatment have a complex interplay with the immune tumor microenvironment (TME). For example, the treatment with 5-fluorouracil (5-FU), often combined with anti-EGFR Abs, results in M1 polarization of the macrophages fostering anti-tumor activity both in vitro and in vivo [62]. In contrast, other chemotherapeutic agents, such as oxaliplatin, remodel TME by inducing Notch signaling and chemo-resistance [62].

In vitro studies revealed that CRC cell lines resistant to cetuximab or the EFGR inhibitor, gefinitib, produced a number of inflammatory cytokines, including *IL1A*, *IL1B*, and *IL8* [55,56]. The expression of these cytokines correlated with the lack of response to EGFR targeting in patient-derived tumor xenografts. Therefore, inflammatory cytokines secreted by resistant CRC cells might be involved in the induction of secondary resistance to EGFR blockade in CRC cells. Impaired cytokine production in peripheral blood mononuclear cells correlates with the response to therapy [55,56]. As a consequence, the inhibition of the production or the action of these cytokines in combination with cetuximab might represent an effective treatment strategy for CRC patients, refractory to anti-EGFR targeting. A recent study showed that CRC patients with high levels of IL1 receptor (IL1R) do not respond to cetuximab treatment, supporting the relevance of IL1 in the resistance to EGFR blockade [56]. The correct dosage of cetuximab and the presence of tumor infiltrating CD8+ T cells are also the key determinants triggering a mechanism known as antibody-dependent cellular cytotoxicity (ADCC). Studies in APC^min/+^ mice models have revealed a novel EGFR-independent oncogenic signal of EGF in the tumor microenvironment [61]. So, strategies that target immune effectors including CD15/FUT4, LY6G6D/F, CD137, CD73, or NK-activating targets may enhance the efficacy of anti-EGFR therapies (Figure 2) [41,42,63,64,65,66,67,68,69,70]. Furthermore, mCRC patients treated with cetuximab plus chemo-therapy show altered cytokine production by the peripheral blood cells after treatment (specifically, an increase in IL-2, IFN-γ, IL-12, and IL-18, and a decrease in IL-4 and IL-10) which correlated with the response to therapy. This suggests that monitoring of the peripheral immune system function can be used as a surrogate marker to predict the treatment-related outcome in these patients (Figure 2) [41,42,61,62]. Notably, EGF produced by macrophages is crucial for the malignant progression of CRC and EGFR inhibition may activate the programmed cell death protein 1 (PD-1/PD-L1) pathway to alter the immune cell function [59]. Apart from the essential roles in metabolism and cell stemness, insulin and EGF are involved in the up-regulation of PD-L1 expression in colon cancer stem cell (CSCs), suggesting that the inhibition of insulin and EGF/EGFR pathways can be considered for cancer immunotherapy, either alone or in combination with PD-1/PD-L1 antibody-based cancer immunotherapy to eliminate CSCs [71]. Besides, immune checkpoint blockade (ICB), immunotherapies aiming at re-activating the T-cell-mediated anti-tumor response, and the adoptive cell transfer (ACT) of natural or gene-engineered ex vivo expanded tumor-specific T cells, have led to unprecedented clinical responses in some tumors. For example, adoptive immunotherapy using T cells engineered to express a chimeric antigen receptor (CAR) specific for the HER2 tumor-associated antigen significantly enhanced CAR T-cell efficacy directed against the HER2 antigen in colorectal cancer in mice models [72]. These data open new perspectives for a combined immunotherapy with anti-epidermal growth factor receptor for patients’ refractory to conventional therapies [60]. It is well-known that the clinical benefits of immune checkpoint blockade are limited in patients with microsatellite instability (MSI) positive advanced colorectal cancer. However, the vast majority of patients with proficient mismatch repair (MMR) or with microsatellite stable (MSS) tumors do not benefit from immunotherapy [21,22,23].

Novel target combinations including MEK inhibitor/checkpoint blockade are under clinical investigation in MSS CRCs. Although just KRAS and NRAS mutations are applied conventionally as an exclusion condition for the use of anti-EGFR monoclonal antibodies, both cell-autonomous and non-cell-autonomous mechanisms, may induce resistance to the treatment. These results provide important mechanistic insights into the dynamic changes occurring in the tumor microenvironment following the treatment with anti-EGFR and highlight the impact of these therapies on the tumor cells and extracellular microenvironment.

## 5. Alternative Non-Genetic Mechanisms That Evade EGFR-Targeted Agents

Emerging evidences raise the possibility of alternative, non-genetic mechanisms contributing to the appearance of cells that can evade EGFR drug treatment [72,73,74,75,76].

A recent study has shown that overexpression of long non-coding RNA MIR100HG and two embedded microRNAs, miR-100 and miR-125b, are linked to cetuximab resistance [74]. These molecules coordinately repressed five Wnt/β-catenin negative regulators, resulting in increased Wnt signaling. Notably, Wnt inhibition in cetuximab-resistant cells restored cetuximab responsiveness identifying a clinically actionable cause of cetuximab resistance [74]. Recently, the combination of EGFR-targeted agents with miRNA therapeutics has attracted attention as a potential strategy to improve treatment efficacy. In addition, miRNAs in circulation have been proposed as non-invasive tools to monitor anti-EGFR therapy response and to predict resistance [76]. However, before considering the incorporation of miRNA in colorectal cancer treatment, it will be crucial to understand their relevance in specific pathological contexts as well as their contribution in the response to multidrug regimens. Expression of the epidermal growth factor ligands amphiregulin (AREG) and epiregulin (EREG) is positively correlated with a response to EGFR-targeted therapies in colorectal cancer. Recently, an intragenic methylation of AREG and EREG genes has been shown to be inversely correlated with their expression in both colorectal cancer cells and human colorectal cancer samples [75]. Retrospective comparison of colorectal cancer patients treated with anti-EGFR revealed that AREG expression is better than *AREG* gene methylation to predict clinical follow-up [68]. Epigenetic mechanisms contribute to regulate the expression of cancer-associated genes, so that inhibition of histone deacetylases (HDACs) can induce cell cycle arrest and differentiation [58]. HDAC-4/EGFR/ERK1/2 signaling has been shown to regulate colonocyte differentiation, supporting the hypothesis that HDAC might cooperate with EGFR signaling to promote colorectal cancer progression [76,77]. However, it remains to be demonstrated whether a therapeutic strategy combining immune or epigenetic effectors with anti-EGFR might be successful to elicit cancer regression in “immunosuppressed” cancer subtypes characterized by a low degree of immune infiltration, while it will render tumors more susceptible to adaptive therapy [77].

## 6. Concluding Remarks

The complex cross-talks among different kinase cascades along with the existing tumor heterogeneity at the genomic and proteomic level and the high probability of mutations, are the cause of resistance to selective EGFR inhibitors and have proved to be a major impediment to achieve the desired results. Currently, only *RAS*-mutation status is used routinely as a negative predictive marker to avoid treatment with anti-EGFR agents in patients with metastatic CRC, but unfortunately a *RAS*-wild-type status does not guarantee a response. Therefore, the major challenge with the use of these biologic therapies is determining predictive biomarkers to optimize patient selection. In addition, it will be essential to find new biomarkers to follow the response to treatment in CRC patients. Advances in molecular biology over the past decade have enabled a better understanding of the development of CRC. A fundamental achievement has been the possibility of using gene-expression profiles to differentiate the subtypes of CRC into prognostic groups, which led to an appreciation of the extensive intratumoral heterogeneity of CRC. CRC subtypes display radically different responses to anti-EGFR therapy independent of the *RAS*-mutation status. The value of clinically relevant mutations can be improved by analyzing circulating plasma DNA and non-tumor cell autonomous mechanisms, which might help to elucidate acquired resistance mechanisms. Available knowledge on the molecular and immunological landscape of colorectal cancer can help to improve the therapeutic management of patients with mCRC. Thus, future trials in molecularly unselected patients will probably not provide clinically relevant data. Novel bio-informatic strategies coupled with high-throughput analysis of cancer material with clinical response data will be essential to successfully complete these next steps and to improve the outcome of patients with this disease.

## Figures and Tables

**Figure 1 cancers-11-01089-f001:**
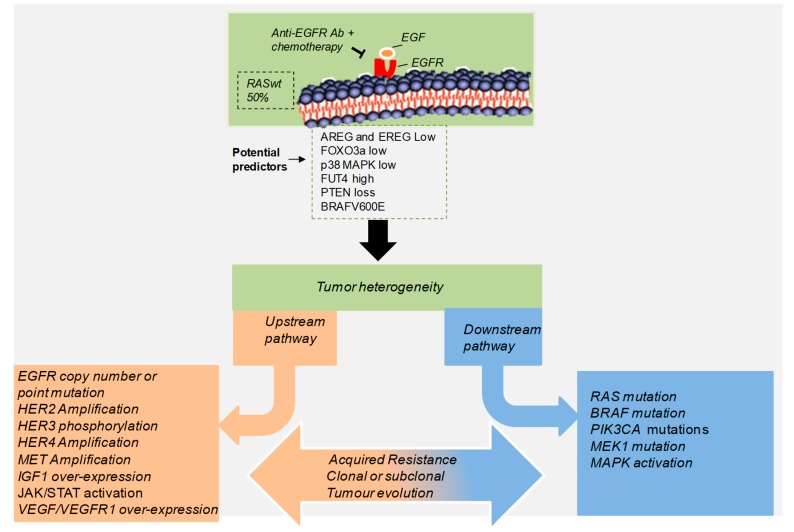
Aberrant genetic alterations implicated in the resistance to anti-EGFR therapy. Anti-EGFR targeted therapy is combined with other cancer treatments in patients with advanced colorectal cancer bearing wild type RAS. While patients respond fairly well initially, in most cases sustained treatment typically results in the failure of the response to treatment and a poor prognosis. This therapy acting as a selection pressure, enables tumor cells to acquire extensive genetic alterations, leading to abnormal activation or amplification of different tyrosine kinase receptors (upstream) and downstream signaling pathways dependent or independent of EGFR signaling. These mechanisms can be expanded through parallel evolution, enabling tumor cells to adapt, while maintaining their intratumoral heterogeneity that contributes to tumorigenesis.

**Figure 2 cancers-11-01089-f002:**
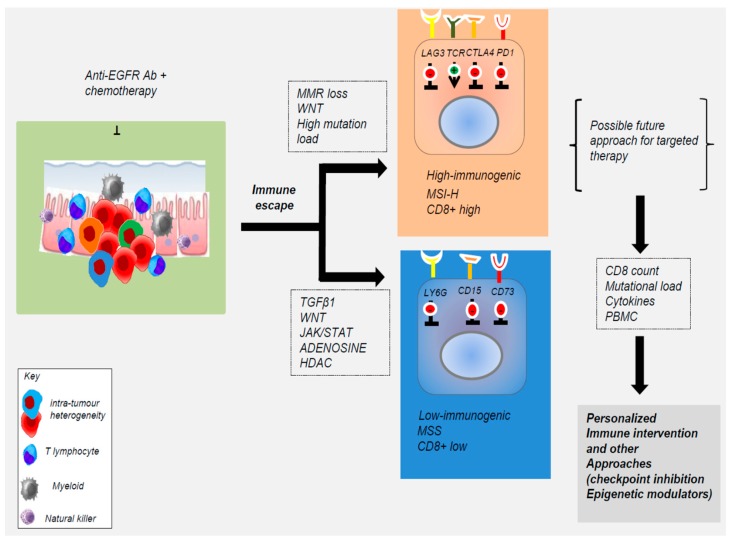
Tumor microenvironment (TME) is a key determinant for the response to anti-EGFR therapy. Tumors are complex adaptive systems owing to the heterogeneous nature of cancer cells and surrounding immune cell populations (T lymphocytes, myeloid cells, natural killer). In the clonal evolution model, from a founder cell, different subclones (represented by different colors) emerge due to genetic and epigenetic alterations resulting in the intratumor heterogeneity. EGFR and its ligands are differentially expressed in the tumor and surrounding immune cell populations. This heterogeneity can significantly affect the effectiveness of targeted therapies. Currently, immunotherapy (Anti-PD1) is restricted to a CRC subgroup harboring loss of mismatch-repair (MMR) proteins. Other CRCs do not meaningfully respond to any traditional immunotherapy approach, including checkpoint blockade, adoptive cell transfer, and vaccination. In the future, pathways and molecules determining the immunological profiling of tumor subtypes might be targeted together with anti-EGFR for therapeutic immune interventions. Abbreviations: microsatellite instable high (MSI-H); *JAK*, Janus kinase; Histone deacetylases (HDAC).

**Table 1 cancers-11-01089-t001:** Current and emerging predictive biomarker for lack of efficacy to anti-epidermal growth factor receptor (EGFR) antibodies therapy in colorectal cancer (CRC) patients.

Gene/Pathway	Genetic Evidence	Population	EGFR Abs	Reference
***Ras/Raf/MEK***				
*KRAS and NRAS*	Mutation	CRC	cetuximab andpanitumumb	[6,7]
*BRAF*	Mutation	*RAS* wild-type CRC	cetuximab and panitumumb	[31,32,33,34]
**Receptors and ligands from EGFR family**				
*Epiregulin (EREG)*	Low expression	*RAS* wild-type and mutant CRC	cetuximab	[35,36,37,38]
*Amphiregulin (AREG)*	Low epression	RAS wild-type and mutant CRC	cetuximab	[35,36,37,38]
EGFR	low copy number		cetuximab	[5]
HER2	Amplification	*RAS/RAF* wild-type mCRC)	cetuximab and panitumumb	[24]
**EGFR downstream**				
PIK3CA	Mutation	*RAS* wild-type CRC	cetuximab	[17,35]
PTEN	loss	*RAS* wild-type CRC	cetuximab	[38,39]
*JAK/STAT3*	Hyper-activated	CRC	cetuximab	[40]
CD15/LY6G6D	High expression	mCRC	cetuximab	[41,42]
**EGFR independent**				
p38 MAPK	Low expression	*RAS* wild-type CRC	cetuximab	[43,44]
FOXO3a	Low expression	*RAS* wild-type CRC	cetuximab	[43,45]

Abbreviations: abs: antibodies.

**Table 2 cancers-11-01089-t002:** Potential genetic polymorphisms predictive for anti-EGFR antibodies efficacy in colorectal cancer reported in literature.

Gene	Polymorphism	Potential Effect	Patient Population
*EGFR*	*C/C* genotype (SNP-994)	less skin toxicity	*RAS* wild type
*EGFR*	T/T genotype (SNP-216)	Better response	*RAS* wild type
*EGFR*	G/G genotype rs1050171 *	predictive of response	*RAS* wild type
UBE2M (involved in EGFR Turnover)	C/C genotype rs895374 *	Predicts short PFS	*RAS* wild type
Fc gamma receptor 3a (*FCGR3A*)	F/F genotype (V158F) **	longer PFS and OS	*RAS* wild type
Toll like receptor 7 (*TLR7*)	G/G genotype rs3853839 *	favorable PFS	*RAS* wild type
killer cell immunoglobulin-like receptor (KIR)	*KIR2DS4* (full-lenght variant)	Predictive of response	*RAS* mutated

* Non-coding region; **, amino acid polymorphism; abbreviations: OS: overall survival; PFS: progression free survival.

**Table 3 cancers-11-01089-t003:** Mechanisms of acquired resistance to anti-EGFR treatment in CRC patients. The main mechanisms involved in the generation of acquired resistance to anti-EGFR therapy are described. They are classified in subgroups according to the pathway(s) and type of alteration.

Gene/Pathway	Genetic Evidence	Study	Reference
*Ras/Raf/MEK* pathway			
*KRAS* and *NRAS*	missense mutations	preclinical and clinical	[6,7]
*BRAF*	missense mutations	clinical and meta-analysis	[31,32]
*MEK1*	missense mutations	preclinical and clinical	[34,35]
**Receptors and ligands from EGFR family**			
*EGFR*	missense mutations	preclinical and clinical	[33]
*HER2*	amplification	preclinical and clinical	[36,37]
HER3/4 ligand	overexpression	preclinical and clinical	[46]
Heregulin	overexpression	clinical	[46]
TGF-α	overexpression	preclinical	[17,18]
**Other tyrosine kinase receptors**			
*MET*	amplification	preclinical and clinical	[5,6,7,8]
IGF1R	overexpression	preclinical	[47]
VEGF/VEGFR	overexpression	preclinical	[4,14,23]
**EGFR downstream signaling**			
PI3K/Akt pathway	hyperactivation	preclinical and clinical	[43,45]
MEK/ERKs pathway	hyperactivation	preclinical and clinical	[34,35]
Foxo 3	upregulation	preclinical and clinical	[43,45]

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
