# Peer review of "Immune Resistance and EGFR Antagonists in Colorectal Cancer"

_cancers, 2019, doi:10.3390/cancers11081089_

Round 1

Reviewer 1 Report

Authors have reviewed the mechanisms of acquired resistance to anti-EGFR therapy on CRC patients. Although the manuscript is quite interesting, I consider that its overall quality can be improved: 

- CRC abbreviation is defined twice in lines 29 and 31. 

- EGFR abbreviation should be defined on Introduction section rather than on line 58.

- Two different abbreviations for 'monoclonal antibodies' can be found along the text, the first one on line 39 (Mab) and the second on line 78 (mAbs). Since Mab has not been longer used, it should be removed and substituted by mAbs. 

- Introduction section should be rewritten in order to clarify the differences between primary and secondary/acquired resistance to anti-EGFR therapies from the beginning. I suggest that lines 40-41 could be changed to: 'Unfortunately, they are only effective in a small percentage of patients [3,5] due to primary or secondary/acquired resistance to this kind of therapy. Alterations in RAS, among other genes, represents the main mechanisms of this resistance [6,7].' In line with this, line 47-48 from 'Notably, even the patients' to 'by a number of causes' should be relocated to define secondary resistance from the beginning. 

- TKIs abbreviation is defined twice in lines 78 and 82. 

- The sections 'HER2 amplification, overexpression of heregulin and EGFR mutations in CRC patients with acquired resistance to anti-EGFR therapy', 'Met amplification or activation in CRC patients with secondary resistance to anti-EGFR treatment' and 'Novel mechanisms of acquired resistance to anti-EGFR therapy' should be engulfed on section 'Acquired resistance to anti-EGFR treatment in CRC patients' since the main topic is defined here. A resume table or another kind of scheme that includes all the mechanisms of acquired resistance should be included at the beginning of this section.

- In lines 155 (twice), 157 and 159, authors have written 'Met' instead of 'MET'.

- Reference 42 appears in red color. 

- In line 188 'wild type' is abbreviated as WT although it has been previously defined as wt on line 91.

- I found line 192-193 confusing and consider that should be reformuled as: 'The down-regulation of both FOXO3a and c-Myc reduced survival and tumor growth'.

- The Abbreviations section lacks many of the abbreviations that can be found along the text e.g. CRC. 

Author Response

Thank you for all the comments as they have allowed us to improve the manuscript.

Here, it is included a point by point response to all the comments.

Q.1 CRC abbreviation is defined twice in lines 29 and 31. 

R.1 This have been corrected. Now, it is only defined in line 29.

Q.2 EGFR abbreviation should be defined on Introduction section rather than on line 58.

R.2 The EGFR abbreviation has been now defined in the Introduction section.

Q.3 Two different abbreviations for 'monoclonal antibodies' can be found along the text, the first one on line 39 (Mab) and the second on line 78 (mAbs). Since Mab has not been longer used, it should be removed and substituted by mAbs. 

R.3 We have modified it, so mAbs is the abbreviation used for monoclonal antibodies in the whole text.

Q.4 - Introduction section should be rewritten in order to clarify the differences between primary and secondary/acquired resistance to anti-EGFR therapies from the beginning. I suggest that lines 40-41 could be changed to: 'Unfortunately, they are only effective in a small percentage of patients [3,5] due to primary or secondary/acquired resistance to this kind of therapy. Alterations in RAS, among other genes, represents the main mechanisms of this resistance [6,7].' In line with this, line 47-48 from 'Notably, even the patients' to 'by a number of causes' should be relocated to define secondary resistance from the beginning. 

R.4. The introduction has been re-written according to the reviewer comments to make it clearer.

Q.5 TKIs abbreviation is defined twice in lines 78 and 82. 

R.5 TKIs abbreviation is now defined only once, just the first time that is mentioned.

Q.6 The sections 'HER2 amplification, overexpression of heregulin and EGFR mutations in CRC patients with acquired resistance to anti-EGFR therapy', 'Met amplification or activation in CRC patients with secondary resistance to anti-EGFR treatment' and 'Novel mechanisms of acquired resistance to anti-EGFR therapy' should be engulfed on section 'Acquired resistance to anti-EGFR treatment in CRC patients' since the main topic is defined here. A resume table or another kind of scheme that includes all the mechanisms of acquired resistance should be included at the beginning of this section.

R.6 According to the reviewer suggestion, we have now included a table (now Table 3) where the main mechanisms of acquired resistance have been described.

Q.7 In lines 155 (twice), 157 and 159, authors have written 'Met' instead of 'MET'.

R.7 It is correct to write Met instead of MET. However, it is true that it should be always written in the same way. So, according to the reviewer suggestion, we have replaced Met by MET.

Q.8 Reference 42 appears in red color. 

R.8 It has changed to black.

Q.9 TKIs abbreviation is defined twice in lines 78 and 82. 

R.9 TKIs abbreviation is now defined only the first time.

Q.10 In line 188 'wild type' is abbreviated as WT although it has been previously defined as wt on line 91.

R.10 This has been changed and it is now written as wt.

Q.11 I found line 192-193 confusing and consider that should be reformuled as: 'The down-regulation of both FOXO3a and c-Myc reduced survival and tumor growth'.

R.11 The sentence has been modified according to the reviewer suggestion.

Q.12- The Abbreviations section lacks many of the abbreviations that can be found along the text e.g. CRC. 

R.12 The reviewer is right, so we have tried to include all the abbreviations in this section.

Reviewer 2 Report

In this review Giordano et al, critically discussed the biological mechanisms involved in the primary and secondary resistance to anti-EGFR agents in colorectal cancer.  The topic is surely of interest and may provide new insight in the definition of predictive molecular markers  that could be used in clinical practice to  individualize and optimize the colorectal cancer treatment. The work is globally good. The authors summarized in a detailed and exhaustive manner the most important results published until to now. The topic is exposed in a very schematic and clear manner. Only some minor comments:

-       The Figure 1 should be made clearer. The “potential predictors” refers only to the white block or also to the orange and blue block? what is the meaning of the question mark? I would specify in the legend. Furthermore, the text also mentions VEGF / VEGFR which is not shown in the figure; verify the completeness of the Figure

-       A summarizing tables with the key details of the studies reported in the text and the relative potential predictors suggested (i.e. therapy setting, study population, in vivo or in vitro models, main findings and so on) will undoubtedly help the readers to easily follow the data presented in the text.

-       In the line 238-239 it is not clear the sentence “This also agrees with the fact that tumors responsive to EGFR antagonist  treatment in vivo are often not sensitive to monoclonal antibody treatment in cell culture when explanted”. It should be better clarified the connection with the data that appear to not support the sentence (i.e. in vitro an effect of anti-EGFR agents on  cytokine has been also reported what is in vivo)

-       In the manuscript there are not cited the role of germ-line genetic polymorphisms in determing the effectiveness of anti-EGFR therapy. For completeness this issue should be mentioned and briefly discussed.

-       Some studies reported the effect of the conventional chemotherapeutic agents routinely used in the CRC treatment (e.g. fluoropyrimidines, oxaliplatin, irinotecan)  on the immune response to therapy with an impact on the clinical outcome. For completeness this issue should be more discussed.

Author Response

Thank you for all the comments as they have allowed us to improve the manuscript.

Here, it is included a point by point response to all the comments.

Thank you for all the comments as they have allowed us to improve the manuscript.

Here, it is included a point by point response to all the comments.

Q.1 The Figure 1 should be made clearer. The “potential predictors” refers only to the white block or also to the orange and blue block? what is the meaning of the question mark? I would specify in the legend. Furthermore, the text also mentions VEGF / VEGFR which is not shown in the figure; verify the completeness of the Figure.

R.1 We have corrected the Figure 1 following the Reviewer’s suggestion in order to make it clearer. The figure legend has also been modified.

Q.2 A summarizing tables with the key details of the studies reported in the text and the relative potential predictors suggested (i.e. therapy setting, study population, in vivo or in vitro models, main findings and so on) will undoubtedly help the readers to easily follow the data presented in the text.

R.2 We have described both current and potential predictive biomarkers for lack of efficacy to anti-EGFR ab therapy in a new Table 1.

Q.3 In the line 238-239 it is not clear the sentence “This also agrees with the fact that tumors responsive to EGFR antagonist treatment in vivo are often not sensitive to monoclonal antibody treatment in cell culture when explanted”. It should be better clarified the connection with the data that appear to not support the sentence (i.e. in vitro an effect of anti-EGFR agents on cytokine has been also reported what is in vivo).

R.3 According to the reviewer comment, we have modified the text in the following way: “This also agrees with the fact that tumors responsive to EGFR antagonist treatment in vivo are often not sensitive to monoclonal antibody treatment in cell culture when explanted, where the immune cells are not present. In fact, inhibition of EGFR signaling in colon cancer cells modulates cytokines and growth factors secretion (e.g. IGF-1) and prevents M1-to-M2 macrophage polarization within the tumor, thereby inhibiting cancer cell growth [51-52]”. We think that now it can be understood that there is a difference between in vivo and in vitro due to the absence of immune cells when working in vitro with CRC cells.

Q.4 In the manuscript there are not cited the role of germ-line genetic polymorphisms in determining the effectiveness of anti-EGFR therapy. For completeness this issue should be mentioned and briefly discussed.

R.4 We taken into account this comment and briefly described the role of genetic polymorphisms at the lines 93-94. We have included some studies (references 25-30) indicating genes and polymorphisms involved in determining the effectiveness of anti-EGFR therapy in a new Table 2.

Q.5 Some studies reported the effect of the conventional chemotherapeutic agents routinely used in the CRC treatment (e.g. fluoropyrimidines, oxaliplatin, irinotecan) on the immune response to therapy with an impact on the clinical outcome. For completeness this issue should be more discussed.

R.5 We have now briefly described at the lines 251-255 (ref. 62) how adjuvant chemotherapeutic agents including 5-fluorouracil (5-FU) and oxaliplatin have opposing effects on the immune microenvironment with an impact on the clinical outcome.
